# Functions of COP1/SPA E3 Ubiquitin Ligase Mediated by MpCRY in the Liverwort *Marchantia polymorpha* under Blue Light

**DOI:** 10.3390/ijms23010158

**Published:** 2021-12-23

**Authors:** Li Zhang, Tianhong Li, Shengzhong Su, Hao Peng, Sudi Li, Ke Li, Luyao Ji, Yaoyun Xing, Junchuan Zhang, Xinglin Du, Mingdi Bian, Yuying Liao, Zhenming Yang, Zecheng Zuo

**Affiliations:** 1Jilin Province Engineering Laboratory of Plant Genetic Improvement, College of Plant Science, Jilin University, Changchun 130062, China; zhang_li18@mails.jlu.edu.cn (L.Z.); litianhong2011@hotmail.com (T.L.); sushengzhong@jlu.edu.cn (S.S.); lisd19@mails.jlu.edu.cn (S.L.); jczhang_gray@163.com (J.Z.); duxinglin2004@163.com (X.D.); bianmd@jlu.edu.cn (M.B.); 2Guangxi Key Laboratory of Veterinary Biotechnology, Guangxi Veterinary Research Institute, Nanning 530001, China; hpeng2006@163.com (H.P.); yuyingliao@163.com (Y.L.); 3Basic Forestry and Proteomics Research Center, Fujian Agriculture and Forestry University, Fuzhou 350002, China; 3200422033@fafu.edu.cn (K.L.); jily199710@163.com (L.J.); xingyaoyun2021@163.com (Y.X.)

**Keywords:** blue light, *Marchantia polymorpha*, cryptochromes, COP1/SPA complex, HY5, ubiquitination, asymmetric growth of thallus

## Abstract

COP1/SPA1 complex in *Arabidopsis* inhibits photomorphogenesis through the ubiquitination of multiple photo-responsive transcription factors in darkness, but such inhibiting function of COP1/SPA1 complex would be suppressed by cryptochromes in blue light. Extensive studies have been conducted on these mechanisms in *Arabidopsis* whereas little attention has been focused on whether another branch of land plants bryophyte utilizes this blue-light regulatory pathway. To study this problem, we conducted a study in the liverwort *Marchantia polymorpha* and obtained a MpSPA knock-out mutant, in which Mp*spa* exhibits the phenotype of an increased percentage of individuals with asymmetrical thallus growth, similar to MpCRY knock-out mutant. We also verified interactions of MpSPA with MpCRY (in a blue light-independent way) and with MpCOP1. Concomitantly, both MpSPA and MpCOP1 could interact with MpHY5, and MpSPA can promote MpCOP1 to ubiquitinate MpHY5 but MpCRY does not regulate the ubiquitination of MpHY5 by MpCOP1/MpSPA complex. These data suggest that COP1/SPA ubiquitinating HY5 is conserved in *Marchantia polymorpha*, but dissimilar to CRY in *Arabidopsis*, MpCRY is not an inhibitor of this process under blue light.

## 1. Introduction

Light is an important energy source and exogenous signal that regulates plant growth and development [1]. To sense and respond to light signals of different wavelengths, plants have evolved specific photoreceptors, including blue-light receptor cryptochromes and red-/far-red light receptor phytochromes [2,3]. Under activation of light, these photoreceptors would immediately inhibit CONSTITUTIVELY PHOTOMORPHOGENIC1/SUPPRESSOR OF PHYA-105 (COP1/SPA) complex [4,5], a light signal repressor, and thus activate the downstream pathway of light signal. In *Arabidopsis*, COP1/SPA complex mediates the ubiquitination and degradation of various light responsive transcription factors to inhibit the light signal regulation in the dark, which include ELONGATED HYPOCOTYL5 (HY5) regulating the de-etiolation of plants [6], CONSTANS (CO) regulating the photoperiodic flowering [7], and PRODUCTION OF ANTHOCYANIN PIGMENT (PAP) regulating the anthocyanin production [8]. COP1 also exists in humans to inhibit tumorigenesis by stabilizing specific transcription factors such as P53 and cJun in a light-independent manner [9,10]. However, SPA proteins are specific to plants and regulate the activity of COP1 in a light-dependent manner [11].

Under blue light (450–500 nm of the visible spectrum), the activity of COP1/SPA is inhibited by photoreceptors including cryptochromes, FKF1 of ZEITLUPE family, and Phytochrome A [5], and such regulation is conducted in five different approaches: (1) blue light induces the nuclear exclusion of COP1, thereby separating COP1 from its target protein [12]; (2) SPA proteins degrade under blue light [13]; (3) blue light activates photoreceptors to bind to SPA1, causing the dissociation of SPA1 and COP1 [14]; (4) COP1 homodimerization would be disrupted by blue light, and thus inhibiting COP1 activity [15]; and (5) blue light activates photoreceptors to inhibit COP1 binding to target proteins through VP domain-mediated competition [16]. Cryptochromes, as typical blue-light receptors, play an important role in blue light inhibition of COP1/SPA activity. In *Arabidopsis*, AtCRY1 is triggered under blue light to induce the nuclear exclusion of AtCOP1, relying on AtSPA1 [17,18]. The activated AtCRY1 would interact with AtSPA1 to inhibit the interaction of AtSPA1 and AtCOP1, leading to the reduced ubiquitination activity of AtCOP1 [14]; while the interaction of AtCRY2 and AtSPA1 would not reduce that of AtSPA1 and AtCOP1 but would strengthen AtCRY2 interacting with AtCOP1 [19]. Previous studies have demonstrated that the VP domain of AtCRY2 binds to the WD40 domain of AtCOP1, thereby inhibiting AtCOP1 binding to target proteins [3,16]. It has been speculated in a recent study that in *Physcomitrella patens*, a kind of non-vascular PpCOP1 ubiquitinates PpHY5 independent of PpSPAs under dark conditions [20], but how blue light regulates COP1/SPA in non-vascular plants has not been studied.

The bryophyte *Marchantia polymorpha* is drawing attention as a new model system in the study of light signals, the genome of which contains typical photoreceptor genes and most of the downstream light signal genes [21]. Previous studies have shown the functions of the single-copy phytochrome ortholog MpPHY, the ultraviolet-B photoreceptor MpUVR8, and the blue light receptors MpPHOT and MpFKF in *M. polymorpha* [22,23,24,25,26]. Unfortunately, few researchers have addressed the functional differences and commonalities of COP1/SPA complex between vascular plant *Arabidopsis thaliana* and the bryophyte *M. polymorpha*. In addition, it is also an important question worth studying whether blue light regulates COP1/SPA activity in *M. polymorpha*.

The *M. polymorpha* genome contains only one *CRY* gene (MpCRY, Mp2g17590), one *SPA* gene (MpSPA, Mp3g25460), and one *COP1* gene (MpCOP1 Mg5g12010). In comparison, there are four paralogs of SPAs in *Arabidopsis* genome and nine paralogs of COP1 in *Physcomitrella*. Therefore, the functional study on COP1/SPA of liverworts is comparably easy to implement. In this study, we obtained Mp*spa1* knockout mutants, and elaborated the function of MpCOP1/MpSPA complex and the effect of blue light receptor MpCRY on its function through the phenotype under blue light and specific biochemical evidence.

## 2. Results

### 2.1. MpSPA Is Associated with MpCRY to Regulate the Thallus Symmetry of M. polymorpha under Blue Light

The *M. polymorpha* genome contains a *SPA* ortholog *MpSPA* (Appendix A), owning three domains as AtSPA1 [27], which are N-terminal kinase-like domain, coiled-coil (CC) domain, and C-terminal WD40 domain (Appendix A). To analyze the function of MpSPA in *M. polymorpha* under blue light, we used CRISPR/Cas9 technology to target the first exon of MpSPA and obtained two gene knockout mutant lines Mp*spa-8* and Mp*spa-11*, among which 5 bases were deleted in Mp*spa-8* and 11 bases deleted in Mp*spa-11* (Figure 1a,c). We also obtained Mp*cry-7* and Mp*cry-8*, two knock-out mutant lines of *MpCRY*. Mp*cry-7* and Mp*cry-8* have a 26-bp deletion and a 1-bp insertion in the 4th exon of MpCRY, respectively (Figure 1b,d). The gemmalings of WT, Mp*spa* mutant, and Mp*cry* mutant were exposed to both blue and red light for 14 days. With continuous illumination of red light, there were no significant growth differences among WT, Mp*spa* and Mp*cry*. Interestingly, under blue light, 57% individuals in Mp*cry-7*, 58% in Mp*cry-8,* 60% in Mp*spa-8,* and 59% in Mp*spa-11* showed asymmetric growth of thallus (the growth from only one side of the gemma, hereafter referred to as ‘asymmetric growth’) under blue light, while less than 5% of individuals in WT showed asymmetric growth (Figure 1f,g).

To verify the regulation of MpCRY to MpSPA under blue light, we used CRISPR/Cas9 technology to obtain two double-knockout mutant lines Mp*cryspa-6* and Mp*cryspa-13* (Figure 1e). We defined the symmetric percentage as the number of the symmetrically growing thallus in 100 thalli. We found that after 14 days of continuous blue light irradiation, the symmetric percentage of Mp*cryspa1-6* and Mp*cryspa1-13* was 47.3% and 48.7%, which is not significantly different from that of Mp*cry* and Mp*spa* single mutant (Figure 1f,g). These results suggested that MpSPA, as MpCRY, could promote the symmetric growth of thallus under blue light, regulating the symmetric growth of thallus in with the identical signaling pathway. However, CRYs inhibit SPA1 in *Arabidopsis* under blue light, while the thallus of Mpcry and Mpspa1 mutants in *M. polymorpha* shared the identical asymmetry ratio under blue light. Therefore, we speculated that MpCRY did not inhibit the activity of MpSPA under blue light.

AtCRY2 protein is rapidly degraded by the 26S proteasome system in response to blue light [28,29,30], whereas AtCRY1 is stable. In addition, AtSPA1 and AtSPA2 also degrade in blue light [13]. In our study, MpCRY degraded in blue light and phosphorylated like *Arabidopsis* CRY2 (Appendix A), while MpSPA accumulated under the same conditions (Appendix A), which is different from AtSPA1 and AtSPA2 in *Arabidopsis thaliana*. Based on the above results, we speculated that MpCRY possesses partial characteristics of AtCRY2, but the stability of MpSPA protein is affected by blue light, which is different from AtSPA1.

### 2.2. MpSPA Interacts with MpCRY in the Nucleus

In *Arabidopsis*, CRY1 and CRY2 interacts with SPA1 in a blue-light-dependent way through C-terminal extension (CCE) domain [14] and N-terminal photolyases homology region (PHR) domain [19], respectively. Similarly, MpCRY possesses N-terminal PHR domain and C-terminal CCE domain as well (Figure 2a). To identify the interaction of MpCRY and MpSPA as well as the interdependent domains achieving such interaction, we first examined whether MpSPA interacts with MpCRY in a blue-light-dependent manner in yeast cells. Yeast two hybrid assay showed that the full-length MpSPA and 607 residues’ lack of the WD40 domain in MpSPA N-terminal fragment could interact with the full-length MpCRY and CCE domain of MpCRY, but the kinase domain, helix-loop-helix domain, and WD40 domain of MpSPA failed to interact with MpCRY (Figure 2b), suggesting that MpCRY interacts with the kinase domain and helix-loop-helix domain of MpSPA, but not necessarily with WD40 domain of MpSPA.

The full-length MpSPA did not interact with the PHR domain of MpCRY, although the N-terminal 607 residues of MpSPA without the WD40 domain interacted with the C-terminal PHR domain of MpCRY in yeast (Figure 2c). Based on these results, we suggested that the interaction-dependent domain of MpCRY and MpSPA1 is the same as that of AtCRY1 and AtSPA1. Moreover, MpCRY and MpSPA generated constitutive interaction in yeast under blue light and dark conditions, which is interestingly different from the blue light-specific interaction of AtCRYs and AtSPA1 in yeast two hybrid assays [14,19]. Consistent with this result, we also found that MpCRY interacted with MpSPA independently of blue light, using immunoprecipitation in HEK293T cells co-expressing Myc-MpCRY and Flag-MpSPA (Figure 2e,f) [31].

In addition, the interaction of MpCRY and MpSPA were detected through Fluorescence Complementation (BiFC) within a transient cell in *Arabidopsis*, the result of which showed that MpCRY and MpSPA interacted in the nucleus (Figure 2d). We then obtained the *M. polymorpha* plants individually expressing and co-expressing *_pro_35S::Citrine-MpSPA* and *_pro_35S::MpCRY-Tdtomatod* to investigate the subcelluar localization of MpSPA and MpCRY in liverworts. It was shown that MpSPA was localized exclusively to the nucleus and formed speckles under blue and dark conditions (Appendix A). MpCRY-Tdtomato is localized to the nucleus and cytoplasm (Appendix A) [32]. Similar to *Arabidopsis* AtCRY1 but different from *Arabidopsis* AtCRY2 which forms photobodies under blue light, individually overexpressed MpCRY-Tdtomato was not able to form photobodies under blue light. In this study, the co-expressed MpCRY and MpSPA were found to locate in the nucleus, in which MpSPA did not constitute speckles in the dark, while MpSPA and MpCRY jointly form photobodies in blue light (Figure 2g,h), showing a high affinity with AtCRY1 which cannot form photobodies when expressed alone (Appendix A), but do generate photobodies when co-expressed with AtSPA1 [33]. Based on the above results, we speculated that MpCRY possessed some properties of AtCRY1 and may form photobodies under blue light to regulate signals when working with MpSPA.

### 2.3. MpSPA Interacts with MpCOP1 in the Nucleus

SPAs interact with COP1 to form a tetramer to enhance the activity of COP1 to ubiquitinate target proteins [34]. MpCOP1 (Mg5g12010) is the ortholog of AtCOP1 in the genome of *M. polymorpha* that also contains a RING-finger domain in the N-terminal binding to ubiquitin conjugating enzyme E2, a coiled-coil (CC) domain in the middle to constitute homodimerization or to form heterodimerization with SPA1, and a WD40 domain in the C-terminal to bind the target proteins and photoreceptors (Figure 3a) [11]. We then used the yeast two-hybrid method to detect the interaction between MpSPA and MpCOP1, finding that full-length MpSPA interacted with full-length MpCOP1 (Figure 3b). Moreover, the coiled-coil domains of MpCOP1 and MpSPA interacted with each other, indicating that, just like AtCOP1-AtSPA1, the interaction of MpCOP1 and MpSPA is performed through their respective coiled-coil domain (Figure 3b). Therefore, the 607 residues at the N-terminal of MpSPA with the deletion of WD40 domain can also interact with the 278 residues at the N-terminal of MpCOP1 without WD40 domain (Figure 3b).

In addition, the yeast co-expressing the kinase domain of MpSPA coupled with the DNA binding domain (BD) of GAL4 and the isolated transcription activation domain (AD) of GAL4 grew on His- medium, indicating the transcriptional activation activity of MpSPA kinase domain (Figure 3b). Using immunoprecipitation assay to co-express Flag-MpSPA and GFP-MpCOP1 in HEK293T cells, we again demonstrated the interaction between MpSPA and MpCOP1 (Figure 3c), which is consistent with our findings that MpSPA and MpCOP1 interacted in the nuclei of the transiently transfected *Arabidopsis* protoplasts with the formation of speckles (Figure 3d,e). Taken together, our results indicate that MpSPA interacts with MpCOP1 in the nucleus, and it is speculated that MpSPA and MpCOP1 in *M. polymorpha* form the E3 ligase complex to ubiquitinate the target protein, as the AtCOP1/AtSPA1 complex. We also tried to generate MpCOP1 knock-out lines by CRISPR/Cas9. As a previous study has shown [34], these lines were lethal (data not shown).

### 2.4. MpCOP1/MpSPA Complex Interacts with MpHY5

HY5 is an important target protein of COP1/SPA ubiquitin ligase in *Arabidopsis thaliana*, which could interact with AtSPA1 and AtCOP1. In this study we focused on MpHY5 (Mp1g16800), the ortholog of AtHY5 in *M. polymorpha*, and aimed to figure out its interaction with MpSPA and MpCOP1. Yeast two-hybrid results showed that the coiled-coil domain of MpSPA interacted directly with the C-terminal of MpHY5, different from that of AtSPA1 and AtHY5 in *Arabidopsis* which requires both the WD40 domain and the coiled-coil domain (Figure 4a) [6]. Moreover, we used immunoprecipitation of HEK293T cells co-expressing GFP-MpHY5 and Flag-MpSPA and found that GFP-MpHY5 and Flag-MpSPA interacted in HEK293T cells in which the interaction of MpHY5 and MpCOP1 similarly appeared (Figure 4c–e). To investigate the ability of MpHY5 to regulate symmetric growth of *M. polymorpha* thallus, we utilized CRISPR/Cas9 knockout mutant Mp*hy5-6* and transgenic *M. polymorpha* (proEF1::Mphy5-Citrine/WT) overexpressing cDNA of MpHY5-Citrine in wild type plants. These plants were exposed to blue light for 14 days and did not exhibit the similar phenotype of Mp*cry* and Mp*spa* that the percentage of thallus with asymmetric growth increases (Figure 4f,g), suggesting that MpHY5 do not regulate the symmetric growth of thallus.

### 2.5. MpSPA Promotes That MpCOP1 Ubiquitinates MpHY5

To verify the participation of MpSPA to MpHY5 degradation, we carried out an in vitro assay [35] to detect the effect of MpSPA on the stability of MpHY5 proteins by incubating purified recombinant His-GFP-MpHY5 proteins with tissue extracts from the Mp*spa* mutant and WT plants. Not surprisingly, the recombinant MpHY5 proteins were much more sensitive to the seedling extract of the WT plants, compared to the seedling extract of the Mp*spa*, and degraded much more quickly (Figure 5a,b). It showed that MpSPA is required for the ubiquitination and degradation of MpHY5 under dark.

SPA1 could promote COP1 activity in ubiquitination of the target proteins in *Arabidopsis* [36]. To verify whether MpSPA possess such ability, we used HEK293T cells to analyze the regulation of MpSPA and MpCOP1 on MpHY5 ubiquitination. The immunoprecipitation-enriched GFP-MpHY5 showed weak ubiquitination when co-expressed with MpCOP1, but had a noticeably strong ubiquitination when co-expressed with MpSPA and MpCOP1 in HEK293T cells (Figure 5c,d). Thus, we wondered whether MpSPA enhancing the ubiquitination of MpHY5 originated from MpSPA promoting the interaction between MpCOP1 and MpHY5. To this end, we carried out co-immunoprecipitation using HEK293T cells co-expressing GFP-MpHY5, Flag-MpSPA, and Myc-MpCOP1. The results showed that MpSPA did not promote the interaction between MpCOP1 and MpHY5 (Figure 4d). In summary, our results indicated that MpCOP1 could ubiquitinate MpHY5, and MpSPA would promote such ubiquitination.

### 2.6. Blue Light Increases MpHY5 Abundance Independently of MpCRY

To investigate whether blue light affects the stability of MpHY5, we prepared the transgenic plants expressing the MpHY5 coding region fused with Citrine open reading frame, driven by the *M. polymorpha* ELONGATION FACTOR1a (Mp-EF1) promoter [37]. We transferred _pro_EF1::MpHY5-Citrine/WT transgenic gemmalings, pre-treated in the dark for 2 days, to blue light for detecting the levels of MpHY5 proteins. As observed in *Arabidopsis* for the AtHY5 protein [14], blue light increases the abundance of MpHY5 proteins (Figure 6a,b). In *Arabidopsis*, blue-light activated AtCRY1 suppresses the AtSPA1–AtCOP1 interaction and AtCOP1-dependent degradation of the transcription factor AtHY5 [14]. To investigate whether MpCRY mediates the accumulation of MpHY5 proteins, we generated the transgenic plants _pro_EF1::MpHY5-Citrine/Mp*cry*. We found that in Mp*cry* mutants, MpHY5 can also accumulate as in the WT plants (Figure 6a,b). This result suggested that the accumulation of MpHY5 is not dependent on MpCRY under blue light and some other blue light receptors may control this process. In *Arabidopsis*, AtCRY1 interacts with AtSPA1 to suppress AtSPA1-AtCOP1 interaction. To test this function of MpCRY, we performed a Co-IP assay through co-expressing GFP-MpSPA, Myc-MpCRY, and Flag-MpCOP in HEK293T cells. Different from AtCRY1, MpCRY1 failed to inhibit the interaction of MpSPA and MpCOP1 (Figure 6c,d). Because both MpCRY and MpHY5 interact with the coiled-coil domain of MpSPA, we speculated that MpCRY can act as the competitive inhibitor of the MpSPA-MpHY5 interaction. To prove this speculation, Flag-MpSPA, Myc-MpCRY, and GFP-MpHY5 were co-expressed in HEK293T cells. The result of the Co-IP assay showed that Myc-MpCRY weakly inhibited the interaction between Flag-MpSPA and GFP-MpHY5 (Figure 6e,f). To further investigate whether MpCRY suppresses the activity of MpCOP1/MpSPA complex in ubiquitination of MpHY5, we conducted an IP assay using HEK293Tcells co-expressing GFP-MpHY5, Flag-MpSPA, Flag-MpCOP1, and Myc-MpCRY and found that MpCRY failed to inhibit the activity of MpCOP1/MpSPA complex in ubiquitination of MpHY5 (Figure 5c,d). In summary, blue light increases MpHY5 abundance independently of MpCRY.

## 3. Discussion

Cryptochromes are conserved flavoprotein receptors identified throughout the biological kingdom with diversified roles in organism development and entrainment of the circadian clock [38,39,40]. Although a myriad of studies have reported a variety of regulatory mechanisms for blue light to regulate plant growth and development, cryptochromes are mainly found to regulate the photoperiod flowering time and hypocotyl elongation in plant lineage [39,41,42]. We have reported a classic regulatory signaling pathway explaining the mechanism of cryptochromes regulating the hypocotyl elongation in *Arabidopsis*, in which cryptochromes inhibit the activity of COP1/SPA1 E3 ubiquitin ligase to regulate HY5 ubiquitination and hypocotyl elongation [14]. Subsequently, the signaling pathway of cryptochrome regulating COP1 complex has been found in various plants [43,44]. Since cryptochromes and their SPA/COP1/HY5 pathway are so important and widely conserved in plant kingdom, in this study, we raised an intriguing question of whether MpCRY and its signaling pathway are conserved or diversified in the bryophyte plant without the flower and the typical hypocotyl, which is different from higher plant species.

Compared to the model plant *Arabidopsis* which possesses homologous genes, e.g., AtSPA1, AtSPA2, AtSPA3, and AtSPA4 [45], the *M. polymorpha* genome contains single MpCRY, MpSPA, MpCOP1, and MpHY5. Without the typical hypocotyl, the Mp*cry* mutant exhibited a novel phenotype differing from the At*cry1cry2* mutant, which decreased the percentage of asymmetric thalli under blue light. Similar to the Mp*cry* mutant, Mp*spa* mutant and Mp*cryspa* double mutant showed the reduction of asymmetric thalli (Figure 1f,g), suggesting that MpSPA and MpCRY are in the same signaling pathway in regulating the symmetric growth of thallus under blue light. Interestingly, although MpCRY and MpSPA are in the same signaling pathway, which is similar to AtCRY1 and AtSPA1, the molecular functions are different from those of *Arabidopsis*. In *Arabidopsis*, AtCRY1 blue light specifically inhibits the function of AtSPA1, thus under blue light, the hypocotyl of the At*cry1* mutant becomes longer [14]. However, in *M. polymorpha*, the interaction between MpSPA and MpCRY was independent of blue light (Figure 2b,c,e,f), which implied that MpCRY transduces its blue light signal via additional proteins or other manners except for protein–protein interaction in *M. polymorpha*. As we reported previously, AtCRY2 forms blue light specific photobodies after blue light irradiation and the light-activated AtCRY2 has also been shown to co-localize in nuclear bodies with AtCOP1 and AtSPA1 [19]. We thus investigate whether MpCRY could form the photobody and even transduce the signal to proteins co-localizing in the same photobody. Notably, MpCRY could not form photobodies under blue light as AtCRY2, but when co-expressing with MpSPA, MpCRY can form photobodies with MpSPA under blue light (Figure 2f). We speculated that under blue light, MpCRY and MpSPA could regulate the symmetrical growth of thalli by forming blue light specific photobodies rather than that AtCRYs blue light specifically interact with SPA1 to regulate the hypocotyl elongation in *Arabidopsis*.

In addition, as we reported previously, AtCRYs mediated blue light to inhibit the function of the AtCOP1/AtSPA1 complex and the ubiquitination of AtHY5 [14]. However, in *M. polymorpha*, MpCRY did not inhibit the interaction of MpCOP1 and MpSPA. Although MpCRY weakly inhibited the interaction between MpSPA and MpHY5 (Figure 6c), it did not affect the ubiquitination of MpCOP1/MpSPA on MpHY5, which suggested that blue light does not inhibit the activity of MpCOP1/MpSPA through MpCRY in *M. polymorpha*, unlike in *Arabidopsis*. This study exhibited the novel phenotype of plant cryptochrome and the mechanism of cryptochrome-related signaling pathway in bryophytes. It further shed a new light on the study for the biological diversity of cryptochrome signaling transduction and even enlightened the mechanism investigation of cryptochrome which may still be obscure in higher plants.

## 4. Materials and Methods

### 4.1. Plant Materials and Growth Conditions

Male and female *M. polymorpha* accessions, Takaragaike-1 (Tak-1) and Takara-gaike 2 (Tak-2) [46], were cultured aseptically on half-strength Gamborg’s B5 medium [47] under continuous white light of 50 to 60 μmol photon m^−2^ s^−1^ at 22 °C. F1 spores were obtained by crossing Tak-2 and Tak-1. For the observation of the phenotypes of gemmalings growing under blue or red light, we used a blue (450 nm) LED illuminator or a red (657 nm) LED illuminator to give the indicated light conditions. Light intensity was measured by LI-250A Light Meter (LI-COR).

### 4.2. HEK-293T Cell Culture and Transfected

HEK-293T cells (ATCC,ATCC^®^CRL-11268TM) were cultured in DMEM (Biological Industries, 06-1055-57-1ACS) supplemented with 10% (*v*/*v*) FBS (Biological Industries, 04-001-1ACS), 100 U/mL penicillin, and 100 mg/mL streptomycin (Hyclone, SV30010) in humidified 5% (*v*/*v*) CO_2_ in air, at 37 ℃. HEK-293T cells were seeded at a density of 3 × 10^5^ cells per well in a six-well plate and transfected using the PEI-max method as previously described [48].

*Myc*, *Flag*, and *GFP* were inserted into pCI (neo) (Promega, E1841, Madison, WI, USA) using the EcoRI restriction site for expressing detectable proteins in HEK-293T cells. The coding sequences for *MpCRY*, *MpSPA*, *MpCOP1*, and *MpHY5* were then cloned into pCI (neo) Myc, pCI (neo) Flag, or pCI (neo) GFP using XbaI and XmaI restriction sites. The sequences of all primers for the constructions of these plasmids are listed in Appendix A.

### 4.3. Generation of Transgenic Lines

To generate the Mp*cry* and Mp*spa* knockout mutant, we used the CRISPR/Cas9 genome editing system [49]. One guide RNA (gRNA) target sequence was designed in the 4th exon of the Mp*CRY* gene and another gRNA target sequence was designed in the first exon of the Mp*SPA*. The annealed oligonucleotides of the gRNA sequence were cloned into the BsaI site of pMpGE_En03 [50]. Using the Gateway LR reaction (Thermo Fisher Scientific), the gRNA expression cassettes were transferred to the pMpGE011 vector [50] to generate pMpGE011_MpCRY and pMpGE011_MpSPA. As described previously [26], Agrobacterium-mediated transformation of F1 spores was performed to generate the Mp*cry*, Mp*spa* mutant lines. Mp*cryspa* double mutants were generated by co-transformation with two kinds of agrobacterium strains respectively containing pMpGE011_MpCRY and pMpGE011_MpSPA. The Mp*hy5* mutant lines were provide by Nakanisi at Kyoto University.

To obtain the overexpression lines of MpCRY-Tdtomato, the coding sequence (CDS) of MpCRY without the stop codon was amplified and cloned into pENTR/D-TOPO vector (cat # K240020, Thermo Fisher Scientific, Waltham, MA, USA). Then, the cloned sequence was transferred to the destination vector pMpGWB130 [51] to generate *_pro_35S::MpCRY-Tdtomato* binary vector, which was transformed into WT (Tak-1) thalli. To obtain the overexpression lines expressing *Citrine-MpSPA*, the CDS of MpSPA was amplified and cloned into pENTR-1A vector (cat # A10462, Thermo Fisher Scientific). The cloned sequence was then transferred to the pMpGWB305 [51] destination vector to generate *_pro_35S::Citrine-MpSPA* binary vector, which was transformed into WT (Tak-1) thalli. To examine the co-location of MpCRY and MpSPA proteins, we generated transgenic lines co-expressing *_pro_35S::MpCRY-Tdtomato* and *_pro_35S::Citrine-MpSPA* by co-transformation with two kinds of agrobacterium strains respectively containing *_pro_35S::MpCRY-Tdtomato* and *_pro_35S::Citrine-MpSPA* binary vectors. To generate the overexpression lines of MpHY5-Citrine, the CDS of MpHY5 without the stop codon was amplified and cloned into pENTR/D-TOPO vector. Using the Gateway LR reaction, the cloned sequence was transferred to the destination vector pMpGWB106 [51] to generate *_pro_35S::MpHY5-Citrine* binary vector and then was transformed into WT thalli and Mp*cry* mutant lines [52]. To generate the overexpression lines of MpCRY-Flag, the CDS of MpCRY without the stop codon was amplified and was transferred to the destination vector pMpGWB310 [51] to generate *_pro_EF::MpCRY-Flag* binary vector and then was transformed into WT thalli. The sequences of all primers used in transgenic plant generation are listed in Appendix A.

### 4.4. Colocalization and Localization

Gemmae were plated on Gamborg’s B5 medium and imbibed in the dark for 3 days. Then, the gemmalings were incubated under blue light (30 μmol m^−2^ s^−1^) or kept in the dark for 6 h. Leica TCS SP8X confocal microscope was used to observe the gemmalings. A 514 nm laser was used for excitation of the fluorescent protein Citrine and a 560 nm laser was used for excitation of the furescent protein Tdtomato.

### 4.5. Phylogenetic Tree Analysis

For the alignment of amino acid sequences, we used the MUSCLE program in Geneious software (Biomatters, Auckland, New Zealand, version 8.1.3) [53]. After sequence alignment, sequence gap and sequences at both ends were manually removed, and conserved region was retained to calculate evolutionary distance. Phylogenetic tree was built by the online program PhyML 3.0 (www.atgc-montpellier.fr/phyml/ (accessed on 21 March 2020)) [54] with JTT model and four substitution rate categories. Its tree searching was started from aBioNJ tree, and the tree was optimized with Subtree Pruning and Regrafting (SPR) topological moves. For statistical analysis of the constructed phylogenetic tree, bootstrapping was carried out by resampling trees 1000 times. The beautification of the resulting tree was done in Geneious software.

### 4.6. Co-Immunoprecipitation in HEK293T Cells

Next, 36–48 h after transfection, the cells were exposed by blue light or kept in the dark for 3 h, washed twice with PBS buffer (cat # SH30256.01, HyClone, Logan, UT, USA), and then digested with TrypLE™ Express (1×) (cat # 12605-028, Gibco, Waltham, MA, USA) at 37 ℃ for 5 min. The cells were harvested for co-immunoprecipitation.

Cells transfected with different plasmid DNA were lysed by in 500 μL Pierce IP Lysis Buffer (cat # 87787, Pierce, Waltham, MA, USA) with 1× EDTA-free Protease Inhibitor Cocktail Tablets (cat # 4693159001, Roche, Waltham, MA, USA) and incubated on ice for 15 min. The mixtures were then centrifuged at 14,000× *g* for 10 min at 4 ℃ to remove cell debris. The supernatant was mixed with 20 μL GFP trap beads or Anti-DDDDk-tag mAb-Magnetic Agarose beads (cat # M185-10, MBL, Beijing, China), incubated with vertical blending at 4 ℃ for 2 h. The beads-protein complex was washed 4 times with washing buffer (20 mM HEPES (pH 7.5), 40 mM KCl, 1 mM EDTA) and denatured by mixing thoroughly with 30 μL 4× Loading buffer and heating at 100 ℃ for 10 min.

### 4.7. Yeast Two-Hybrid Assay

Experiments using the yeast two-hybrid system are according to the manufacturer’s instructions (Matchmaker user’s manual, Clontech, CA, USA). The coding sequences of MpCRY, MpSPA, and different domains of MpCRY and MpSPA were fused in-frame with the GAL4 DNA-binding domain (BD) of the bait vector pGBKT7 using EcoRI (cat # 630489, Clontech, CA, USA). The coding sequences of MpSPA, MpCOP1, MpHY5, and different domains of these proteins were fused in-frame with the GAL4 transcription-activation domain (AD) of the prey vector pGADT7 (cat # 630442, Clontech, CA, USA). The bait plasmid and the prey plasmid were co-transformed into the yeast strain Y2HGold. All sequence of primers for the constructions of these bait plasmids and prey plasmids were listed in Appendix A.

The protein–protein interaction was analyzed using the histidine auxotrophy assay. Yeast colonies selected on plates (SD-LW) were patched in duplicate onto His− and His+ plates. Then, one duplicate was grown under blue light (30 μmol m^−2^ s^−1^) at 30 ℃ for 2–3 days. The second duplicate was kept in dark in the same condition.

### 4.8. Bimolecular Fluorescence Complementation (BiFC) Assay

Isolation and purification methods for Arabidopsis protoplasts are based on “Tape-*Arabidopsis*-Sandwich” [55] with some modifications. Leaves of 3-week-old *Arabidopsis* plants in LD (16 h light, 8 h dark) were peeled away from the lower epidermal surface using breathable tape (cat # 1530C-0, 3 M Micropore™, St. Paul, MN, USA) and colorful tape (VBWINTAPE). The remaining leaves were transferred to the enzyme solution (20 mM MES pH 5.7, 1.5% (*w*/*v*) cellulase R10, 0.4% (*w*/*v*) macerozyme R10, 0.4 M mannitol, 20 mM KCl and 10 mM CaCl). After 2 h digestion at room temperature, the solution was filtered through a 75-micron nylon mesh (Cat # 475855-1R, Calbiotecm^®^, Billerica, MA, USA) and centrifugated at 100 *g*, 4 ℃ for 2 min to pellet the protoplasts. Then the protoplasts were washed with W5 buffer (2 mM MES pH 5.7, 154 mM NaCl, 125 mM CaCl_2_, 5 mM KCl, 5 mM D-glucose) twice and incubated on ice for 30 min. For PEG-Ca^2+^ protoplast transformation, protoplasts were washed with ice-cold MMG buffer (4 mM MES pH 5.7, 0.4 M mannitol, 15 mM MgCl_2_) twice and resuspended with 100 μL MMG buffer per sample. The indicate volumes of plasmids were added and mixed with protoplasts. The mixture was then mixed with 110 μL PEG-Ca^2+^ buffer (40% (*w*/*v*) PEG4000, 0.2 M mannitol, 100 mM CaCl_2_) and kept at room temperature for 5 min, washed twice with W5 buffer at room temperature. The protoplasts were finally resuspended with 1 mL W5 buffer and transferred into a 6-well plate and incubated for at least 12 h in dark. For MpCOP1-MpSPA BIFC assay, the fluorescence images were captured using an Olympus FV1000 confocal laser scanning microscope. Image analyses were performed using the FV10-ASW 3.0 Software and processed with Adobe Photoshop. For the other BIFC assays, all pictures were captured by a fluorescence microscope (Axio Observer A1, Zeiss, Oberkochen, Baden-Wuberg, Germany). Image analyses were performed using the Zen software (Zeiss) and processed with Adobe Photoshop. All plasmids used in the BIFC assays were provide by Kohchi group at Kyoto University.

### 4.9. Ubiquitination Assays in HEK293T Cells

Ubiquitination assays in HEK293T cells were performed as previously described [56]. HEK293T cells were transiently transfected with 5 μg of the combinations of constructs for 36 h. Ubiquitinated GFP-MpHY5 was detected by IP with GFP trap beads at 4 ℃ for 2 h. The pellets were washed 4 times with washing buffer (20 mM HEPES (pH 7.5), 40 mM KCl, 1 mM EDTA) and denatured by mixing thoroughly with 30 μL 4× Loading buffer and heating at 100 ℃ for 10 min.

### 4.10. Antibodies Used for Immunoblotting

The following primary antibodies used in this study are commercially available: anti-Myc (cat # M192-3, MBL), anti-GFP (cat # 598, MBL), anti-Flag (cat # M185-3L, MBL), anti-HSP (cat # AbM51099-31-PU, Beijing Protein Innovation, Beijing, China), anti-Ubiquitin (cat # 14-6078-82, Invitrogen, Waltham, MA, USA), and anti-Citrine (cat # 632381, Takara, Tokyo, Japan).

### 4.11. Statistical Analysis

All data were analyzed using GraphPad Prism (version 8.0.2) and statistically ana lyzed by two-tailed Student’s *t*-test. Signals on Western blots were quantified using ImageJ software. The partition ratio was calculated based on the previous research [57]. The images of indicated time point were collected and analyzed by Image J. The intensity of particles and the intensity of background were calculated, and the final ratio was fit to the formula:(1)$$Partition\;ratio=\frac{\sum_{i=0}^{n}C\left(i\right)}{n\cdot N\;}$$
where the *C*(*i*) represents the intensity of each condensate; *n* represents the condensate number of each nucleus, and *N* represents the mean intensity of nucleus.

## Figures and Tables

**Figure 1 ijms-23-00158-f001:**
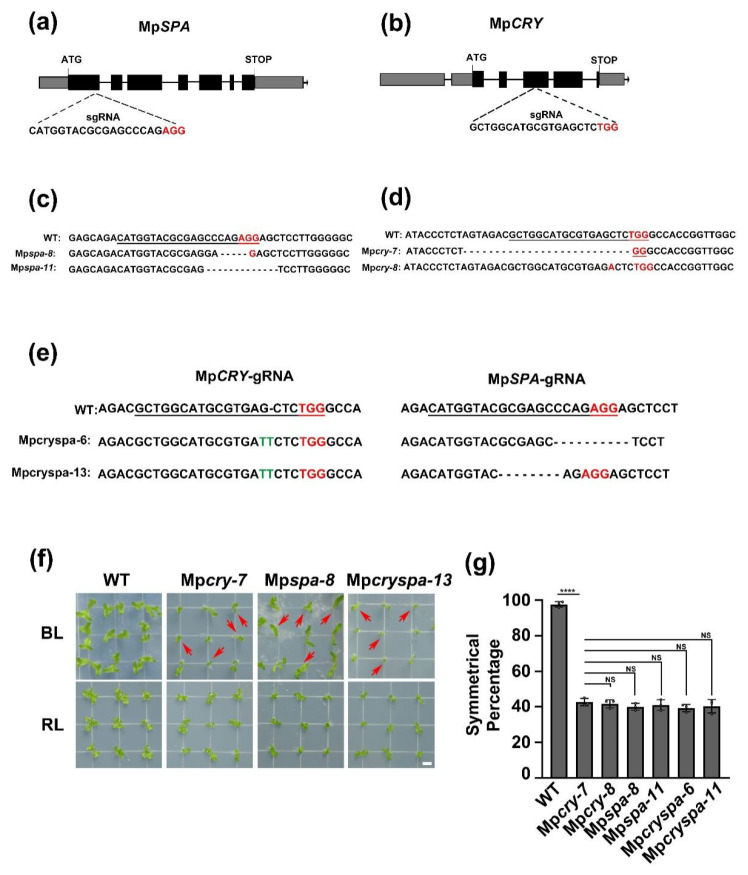
MpSPA is associated with MpCRY to regulate the thallus symmetry of *M. polymorpha* under blue light. (**a**,**b**) Schematic diagram of the structures of the MpCRY gene (**a**), the MpSPA gene (**b**) and the target sequences of CRISPR/Cas9 genome editing. Black boxes indicate CDS regions. Exons contain grey boxes and black boxes. Black lines indicate introns. (**c**,**d**) Mp*cry* (**c**) and Mp*spa* (**d**) mutants detected by sequence analysis. The PAM (protospacer adjacent motif) sequences and bases inserted at the target site were highlighted in red, and the target sequences were underlined in black. Dashes indicate deleted bases. (**e**) Patterns of DNA mutations detected in Mp*cryspa* double mutant lines. The PAM sequences were highlighted in red, and the target sequences were underlined in black. Dashes indicate deleted bases. Substituted bases were highlighted in green. (**f**) Photographs of gemmalings under BL (30 μmol m^−2^ s^−1^) or RL (30 μmol m^−2^ s^−1^) for 14 days. Bar = 5 mm. The red arrows represent individuals with asymmetric growth of thallus. (**g**) The symmetric percentage represents the percentage of plants showing symmetric growth under BL. The experiments were performed as in (**f**). Data are presented as mean ± SD (*n* = 3 biological statistics). More than 100 gemmalings were used for one count. Student’s *t* test: **** *p* < 0.001, no significant difference (NS).

**Figure 2 ijms-23-00158-f002:**
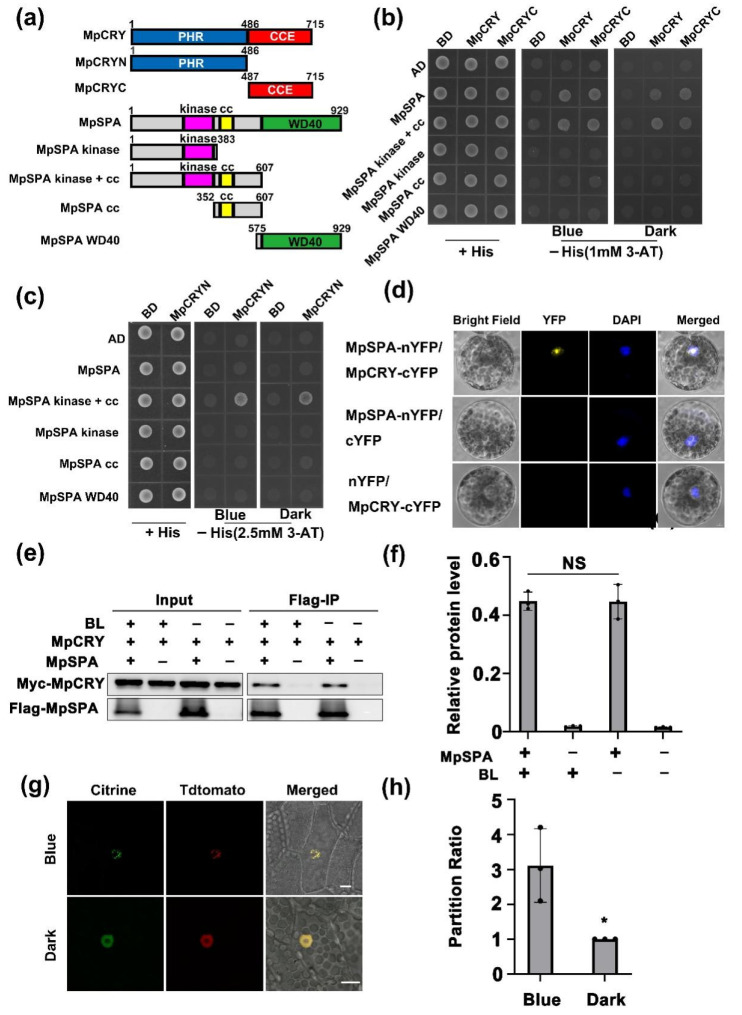
MpSPA interacts with MpCRY in the nucleus. (**a**) Diagrams depicting the linear structures of MpSPA, MpCRY, and their truncate fragments for yeast two hybrid assays. (**b**) Histidine auxotrophy assays showing the interactions between MpCRY and MpSPA, MpCRYC and MpSPA, MpCRY and MpSPA Kinase + CC, or MpCRYC and MpSPA Kinase + CC. Yeast cells containing plasmids encoding the indicated proteins were grown on medium in the presence (+His) or absence (−His) of histidine and supplemented with 1 mM 3-amino-1,2,4,-triazole (3-AT), under blue light (Blue, 30 µmol m^−2^ s ^−1^) or in the dark (Dark) for 3 days. (**c**) Histidine auxotrophy assays showing the interaction between MpCRYN and MpSPA kinase + CC, and the lack of interaction between MpCRYN and MpSPA. The experiments were performed as in (**b**). The medium absence of histidine was supplemented with 2.5 mM 3-AT. (**d**) A BIFC assay in *Arabidopsis* protoplasts showing that MpCRY interacts with MpSPA in the nucleus. Yellow indicates a positive interaction signal, blue indicates signals from 4′,6-diamidino-2-phenylindole (DAPI). Bars = 5 μm. (**e**) Co-immunoprecipitation (Co-IP) assay showing the interaction of MpCRY and MpSPA in HEK-293T cells. The cells were treated with blue light (+BL; 30 μmol m^−2^ s^−1^) for 3 h or kept in the dark (−BL). The immunoprecipitation signals were probed by anti-Myc (MpCRY) or anti-Flag (MpSPA), respectively. (**f**) Quantitative assay of Myc-MpCRY protein levels (IP/Input) from three biological repeats with a representative result shown in (**e**). Data are presented mean ± SD (*n* = 3 biological statistics). Student’s *t* test: no significant difference (NS). (**g**) Colocalization of MpCRY and MpSPA in *M. polymorpha*. Gemmae co-expressing Citrine-MpSPA and MpCRY-Tdtomato were plated on Gamborg’s B5 medium and imbibed in the dark for 3 days. Then, the gemmalings were incubated under BL (30 μmol m^−2^ s^−1^) or kept in the dark for 6 h. Leica TCS SP8X confocal microscope was used to observe the gemmalings. Bar = 10 μm. (**h**) Quantification of partition ratio of Citrine signals of Citrine-MpSPA proteins from the assay shown in (**g**). Data are presented mean ± SD (*n* = 3 biological statistics). Student’s *t* test: * *p* < 0.05.

**Figure 3 ijms-23-00158-f003:**
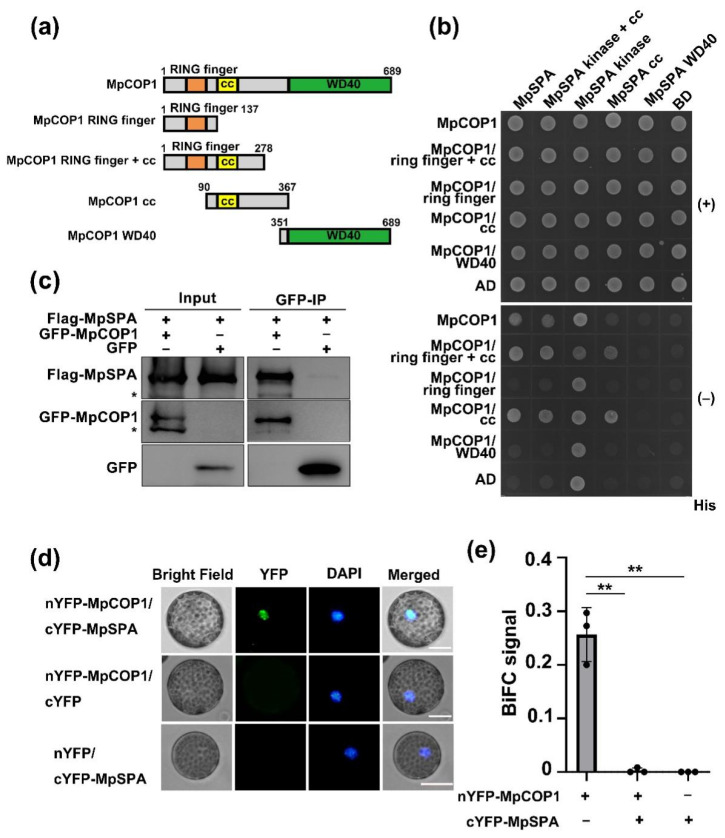
MpSPA interacts with MpCOP1 in the nucleus. (**a**) Diagrams depicting the linear structures of MpCOP1 and its truncate fragments for yeast two hybrid assays. (**b**) Histidine auxotrophy assays showing the interactions between MpCOP1 and MpSPA. Yeast cells containing plasmids encoding the indicated proteins were grown on medium in the presence (+His) or absence (−His) of histidine in the dark for 3 days. (**c**) Co-IP assay showing the interaction of MpCOP1 and MpSPA in HEK-293T cells. The immunoprecipitation signals were probed by anti-GFP (MpCOP1) or anti-Flag (MpSPA), respectively. Asterisks indicate non-specific bands. (**d**) A BIFC assay in *Arabidopsis* protoplasts showing that MpCOP1 interacts with MpSPA in the nucleus. (**e**) BiFC signal showing the ratio of the number of protoplasts with YFP fluorescence signal to the total number of protoplasts in a field of view. The experiments were performed as in (**d**). Data are presented as mean ± SD (*n* = 3 fields of view). Student’s *t* test: ** *p* < 0.01.

**Figure 4 ijms-23-00158-f004:**
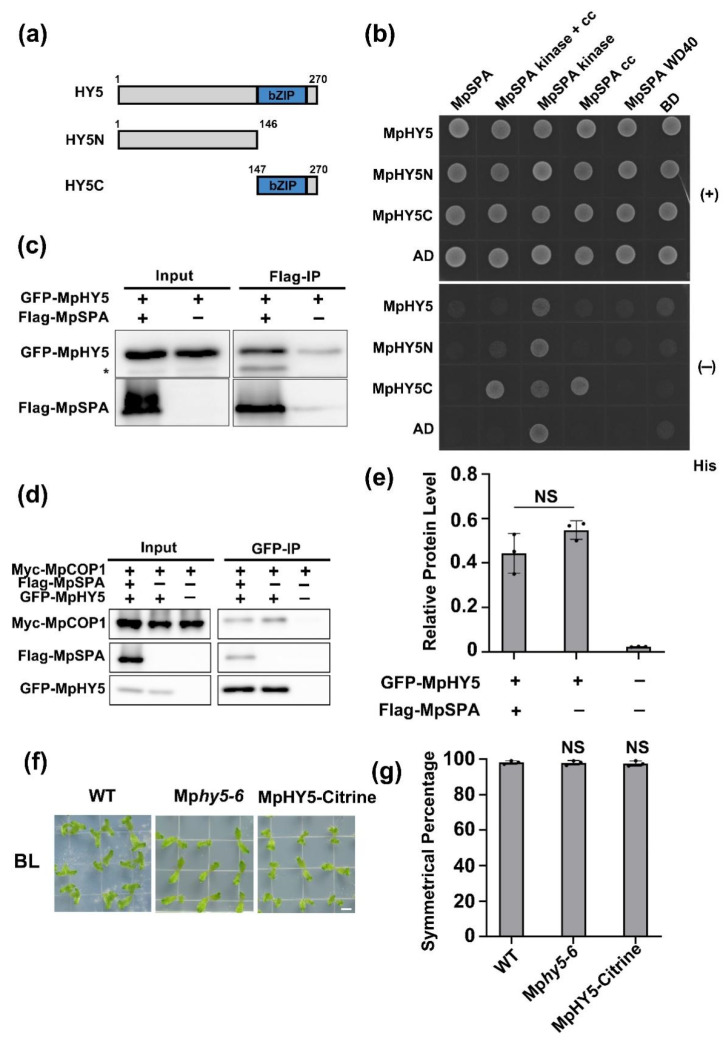
MpCOP1/MpSPA complex interacts with MpHY5. (**a**) Diagrams depicting the linear structures of MpHY5 and its truncate fragments for yeast two hybrid assays. (**b**) Histidine auxotrophy assays showing the interactions between MpSPA CC and MpHY5C. The experiment was performed as Figure 3b. (**c**) Co-IP assay showing the interaction of MpSPA and MpHY5 in HEK-293T cells. The immunoprecipitation signals were probed by anti-GFP (MpHY5) or anti-Flag (MpSPA), respectively. Asterisks indicate non-specific bands (**d**) Co-IP assay showing the interaction of MpCOP1 and MpHY5 in HEK-293T cells, not effected by MpSPA. The immunoprecipitation signals were probed by anti-GFP (MpHY5), anti-Flag (MpSPA), or anti-Myc (MpCOP1), respectively. (**e**) Quantitative assay of Myc-MpCOP1 protein levels (IP/Input) from three biological repeats with a representative result shown in (**e**). Data are presented as mean ± SD (*n* = 3 biological statistics). Student’s *t* test: no significant difference (NS). (**f**) Photographs of gemmalings under BL (30 μmol m^−2^ s^−1^) for 14 days. Bar = 5 mm. (**g**) The symmetric percentage represents the percentage of plants showing symmetric growth under BL. The experiments were performed as in (**f**). Data are presented as mean ± SD (*n* = 3 biological statistics). More than 100 gemmalings were used for one count. Student’s *t* test: no significant difference (NS).

**Figure 5 ijms-23-00158-f005:**
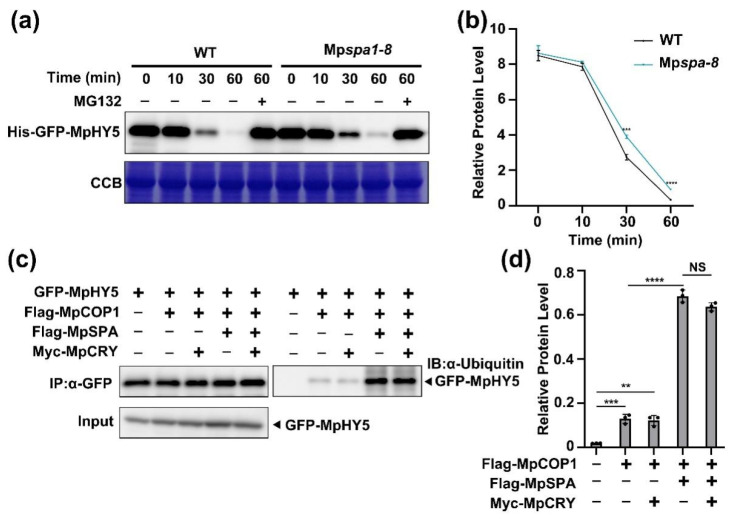
MpSPA promotes that MpCOP1 ubiquitinates MpHY5. (**a**) Assay of GFP-MpHY5 protein stability in vitro. His-GFP-MpHY5 proteins were purified from *E. coli* and incubated with the total extracts of Mp*cry* or WT seedlings in the presence or absence of 50 mM MG132. His-GFP-MpHY5 proteins were probed by anti-GFP on an immunoblot (upper panel). CBB (Coomassie brilliant) blue staining is shown in the lower panel. (**b**) Quantitative assay of His-GFP-MpHY5 protein levels (His-GFP-HY5/CCB) from three biological repeats with a representative result shown in (**a**). Data are presented as mean ± SD (*n* = 3 biological statistics). Student’s *t* test: *** *p* < 0.001. (**c**) Effect of Myc-MpCRY and Flag-MpSPA on the ubiquitination of GFP-HY5 by Flag-MpCOP1 in HEK293T cells. Total proteins were extracted from HEK293T cells transfected with indicated plasmid for Co-IP with GFP trap beads. Proteins were analyzed by immunoblotting with anti-GFP and anti-ubiquitin antibodies. (**d**) Quantitative assay of GFP-MpHY5 protein levels (α-ubiquitin/IP) from three biological repeats with a representative result shown in (**c**). Data are presented as mean ± SD (*n* = 3 biological statistics). Student’s *t* test: ** *p* < 0.01, *** *p* < 0.001, **** *p <* 0.0001, no significant difference (NS).

**Figure 6 ijms-23-00158-f006:**
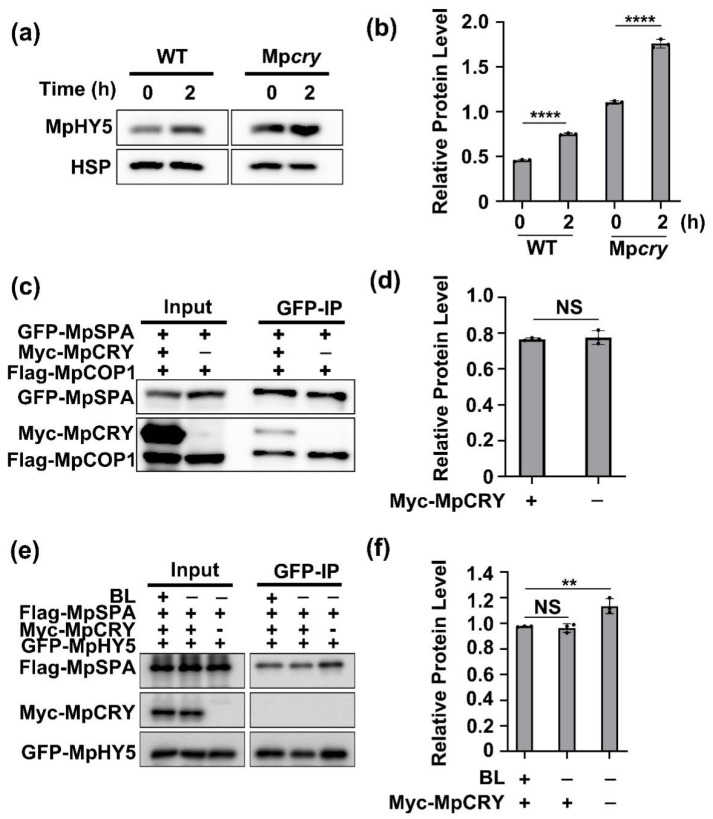
Blue light increases MpHY5 abundance independently of MpCRY. (**a**) Indicated gemmae were plated on Gamborg’s B5 medium for 12 days and then treated in the dark for 2 days. After incubated in blue light (30 μmol m^−2^ s^−^^1^), the levels of HY5-Citrine proteins in extracts from WT and Mp*cry* were detected by immunoblot. HSP protein was used as the loading control. (**b**) Quantitative assay of HY5-Citrine protein levels (HY5-Citrine/HSP) from three biological repeats with a representative result shown in (**a**). Data are presented as mean ± SD (*n* = 3 biological statistics). Student’s *t* test: **** *p* < 0.0001. (**c**) Co-IP assay showing that Myc-MpCRY is ineffective in the interaction between GFP-MpSPA and Flag-MpCOP1 in HEK-293T cells. The immunoprecipitation signals were probed by anti-GFP (MpSPA), anti-Flag (MpCOP1), or anti-Myc (MpCRY), respectively. (**d**) Quantitative assay of Flag-MpCOP1 protein levels (IP/Input) from three biological repeats with a representative result shown in (**c**). Data are presented as mean ± SD (*n* = 3 biological statistics). Student’s *t* test: no significant difference (NS). (**e**) Co-IP assay showing that Myc-MpCRY weakly inhibits the interaction between Flag-MpSPA and GFP-MpHY5. The immunoprecipitation signals were probed by anti-GFP (MpHY5), anti-Flag (MpSPA), or anti-Myc (MpCRY), respectively. (**f**) Quantitative assay of Flag-MpCOP1 protein levels (IP/Input) from three biological repeats with a representative result shown in (**e**). Data are presented as mean ± SD (*n* = 3 biological statistics). Student’s *t* test: ** *p* < 0.01, no significant difference (NS).

## Data Availability

Not applicable.

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
