# Peer review of "Functions of COP1/SPA E3 Ubiquitin Ligase Mediated by MpCRY in the Liverwort *Marchantia polymorpha* under Blue Light"

_ijms, 2021, doi:10.3390/ijms23010158_

Round 1
Reviewer 1 Report
This is a solid paper on a subject the authors have considerable experience handling. A few minor comments:
What wavelength (or range) is meant by "blue" light?
Line 355: I recommend writing out what PPK1 is for novice readers.
Reviewer 2 Report
I checked your manuscript and described comments below.
Superior point
- This article suggest that COP1/SPA1 ubiquitinating HY5 is conserved in Arabidopsis.
- The data in this paper clarified MpCRY is not an inhibitor of this process under blue light.
Minor problems.
Gene symbols etc. must be capitalized in references titles.
I wrote below about the parts that need to be fixed.
Ref. 4 and Ref. 5, cop1/spa e3 ubiquitin ligase -> CPO1/SPA3 E3 ligase
Ref. 8, e3 ubiquitin ligase cop1/spa -> E3 ubiquitin ligase COP1/SPA
Ref. 8, pap1 and pap2 ->PAP1 and PAP2
Ref.9, Cop1 -> COP1
Ref. 10, cop1 -> COP1
Ref. 11, e3 ubiquitin ligase cop1/spa -> E3 ubiquitin ligase COP1/SPA
Ref. 12, cop1 -> COP1
Ref. 13, cop1 -> COP1
Ref. 13, spa2 ->SPA2
Ref. 14, spa1 to suppress cop1 -> SPA1 to suppress COP1
Ref. 15, f-box protein fkf1 -> F-box protein FKF1
Ref. 15, cop1 -> COP1
Ref. 16, cop1 -> COP1
Ref. 17 cop1/spa -> COP1/SPA
Ref. 18, gus-cop1 -> GUS-COP1
Ref. 19, cry2 with spa1 regulates cop1 -> CRY2 with SPA1 regulates COP1
Ref. 20, spa -> SPA
Ref. 25, uvr8 -> UVR8
Ref. 27, Spa1, a wd-repeat protein -> SPA1, a WD-repeat protein
Ref. 29, cry2 -> CRY2
Ref. 30, cry2 -> CRY2
Ref. 31, hek239t -> HEK239T
Ref. 34, cop1 -> COP1
Ref. 34, spa -> SPA
Ref. 35, e3 ligases -> E3 ligases
Ref. 36, Laf1 ->LAF1
Ref. 36, cop1 -> COP1
Ref. 37, mpef1α -> MPEF1α
Ref. 37, camv35 -> CAMV35
Ref. 40, spa1 -> SPA1
Ref. 43, cry2 -> CRY2
Ref. 51, e3 ligases -> E3 ligases
Ref. 51. uv-b -> UV-B
Reviewer 3 Report
Paper "Functions of COP1/SPA E3 ubiquitin ligase mediated by MpCRY in the liverwort Marchantia polymorpha under blue light" is very interesting.
Authors obtained Mpspa1 knockout mutants, and elaborated the function of MpCOP1/MpSPA1 complex and the effect of blue light receptor MpCRY on its function through the phenotype under blue light and specific biochemical evidence.
Quality of all Figures is good.
Paper needs statistical analysis. Authors presented large number of results but lack of testing of significant these data.
Paper needs major revision.
Round 2
Reviewer 3 Report
Thank you for your corrections. Now, is ok.